# The Study of Optimal Adsorption Conditions of Phosphate on Fe-Modified Biochar by Response Surface Methodology

**DOI:** 10.3390/molecules28052323

**Published:** 2023-03-02

**Authors:** Jing Qian, Xiaoyu Zhou, Qingsong Cai, Jinjin Zhao, Xianhuai Huang

**Affiliations:** 1School of Environment and Energy Engineering, Anhui JianZhu University, Hefei 230601, China; 2Anhui Provincial Key Laboratory of Environmental Pollution Control and Resource Reuse, Hefei 230061, China; 3Plant Protection & Quarantine and Tillage & Fertilizer Management Station of Huzhou, Huzhou 313000, China

**Keywords:** phosphate removal, Fe-modified biochar, response surface method, optimization, adsorption–desorption

## Abstract

A batch of Fe-modified biochars MS (for soybean straw), MR (for rape straw), and MP (for peanut shell) were prepared by impregnating biochars pyrolyzed from three different raw biomass materials, i.e., peanut shell, soybean straw, and rape straw, with FeCl_3_ solution in different Fe/C impregnation ratios (0, 0.112, 0.224, 0.448, 0.560, 0.672, and 0.896) in this research. Their characteristics (pH, porosities, surface morphologies, crystal structures, and interfacial chemical behaviors) and phosphate adsorption capacities and mechanisms were evaluated. The optimization of their phosphate removal efficiency (Y%) was analyzed using the response surface method. Our results indicated that MR, MP, and MS showed their best phosphate adsorption capacity at Fe/C ratios of 0.672, 0.672, and 0.560, respectively. Rapid phosphate removal was observed within the first few minutes and the equilibrium was attained by 12 h in all treatment. The optimal conditions for phosphorus removal were pH = 7.0, initial phosphate concentration = 132.64 mg L^−1^, and ambient temperature = 25 °C, where the Y% values were 97.76, 90.23, and 86.23% of MS, MP, and MR, respectively. Among the three biochars, the maximum phosphate removal efficiency determined was 97.80%. The phosphate adsorption process of three modified biochars followed a pseudo-second-order adsorption kinetic model, indicating monolayer adsorption based on electrostatic adsorption or ion exchange. Thus, this study clarified the mechanism of phosphate adsorption by three Fe-modified biochar composites, which present as low-cost soil conditioners for rapid and sustainable phosphate removal.

## 1. Introduction

Phosphorus, an essential nutrient element for plant growth, is often over-fertilized in soil, which usually leads to eutrophication in aquatic environments by runoff or subsurface drainage. Previous studies predominantly focused on the rational application of nitrogen fertilizers and minimizing nitrogen loss to water bodies, while phosphorus pollution in water received insufficient attention. Owing to the excessive application of phosphorus-based fertilizers in agricultural production, the phosphorus in the soil has exceeded the optimal level for plant growth. Thus, agriculture has become a significant phosphorus pollution source. Additionally, phosphorus discharge from industrial wastewater is also another important factor that promotes eutrophication in water environments. Excessive amounts of phosphorus in water bodies makes the water unfit for entertainment, drinking, and industrial purposes [1]. Research into excessive environmental phosphorus removal and recycling methods is attracting more and more attention. Presently, there are several technologies for phosphorus removal in wastewater, including chemical precipitation [2], biofilm processes [3], redox processes [4], and adsorbent adsorption. Among these methods, biochars as adsorbents for adsorption are considered as one of the most promising absorbent materials, owing to the low cost, high efficiency, and environmental friendliness, scalability, and sustainability.

Biochars are carbon-rich materials produced via biomass pyrolysis with limited supply of oxygen and at a relatively high temperature (generally 300–700 °C). Biochars are low cost, given that they are usually obtained from waste biomass materials, such as corn straw, rape straw, wood, manure, sewage sludge, and bamboo [5,6,7,8,9]. Biochars have great potential in sewage and wastewater treatment and soil improvement through phosphorus recovery and environmental pollution remediation, due to their unique physical and chemical properties [10,11]. Generally, biochars have low solubility and high melting points; carry abundant organic functional groups, such as phenolic hydroxyl groups, aliphatic double bonds, and aromatic rings; and show unique structural characteristics, e.g., high porosity and regular tubular structure, which lead to their excellent adsorption capacities and stabilities [12,13].

Rape is one of the main crops for edible oil production in China. According to the Chinese Ministry of Agriculture, the rapeseed cultivation area in China is approximately 6.55 million hectares, and the output accounts for 51% of the national oil-producing crops. In 2018, the annual rapeseed output was 1.33 million tons, producing approximately 3.97 million tons of rape straw. However, the actual utilization efficiency of the waste biomass was less than 25% over the past year [14]. As the largest peanut cultivating country in the world, Chinese annual peanut output accounts for more than 35% of the global total amount. According to the U.S. Department of Agriculture, in 2019, global peanut yield was 45.44 million tons, with China contributing 17.5 million tons, i.e., 38.5% of the total amount. Meanwhile, large quantities of waste peanut shells are produced [15]. Similarly, the output of soybean straw in China is also very high. In 2020, approximately 31.36 million tons of soybean straw will be produced in China [16]. Thus, the use of waste biomass for biochar production not only ensures agricultural waste prevention, but will also lead to the prevention of secondary environmental issues once applied to environmental remediation.

Biochars produced from cellulose- and lignin-enriched biomass materials exhibit differences in adsorption capacity. Some biochars contain acidic functional groups on their surface, such as straw biochar, which would be suitable for cation adsorption, while it has little ability for anion exchange. To improve the adsorption performance of above-mentioned biochars, several metal elements including iron, aluminum, calcium, and magnesium, are used for biochar modification [17]. For example, Fe-modified biochars (Fe-B) are obtained via pyrolysis of raw materials pretreated with FeCl_3_, resulting a combination of adsorption capacity of biochar and reduction and complexing capacity of Fe [18]. Compared with other technologies (porous molecular sieves, magnetic nanomaterials) for adsorbent production, Fe-B as adsorbent is cost-effective, scalable, and sustainable in engineering applications.

The removal of P from water using biochars is a promising method, and has been widely studied in recent years. Biochar modification, especially metal modification, can remarkably improve its adsorption performance to phosphorus in water. Li et al. [19] used lanthanum (La)-modified sludge through impregnation–co-precipitation to remove phosphate from water, and the results showed La-coated biochar at a pyrolysis temperature of 600 °C had the highest phosphate adsorption and lowest heavy metal leaching potential. Yin et al. [20] used two types of metals, Mg and Ca, to modify the biochar structure in aqueous solution to absorb phosphorus; the experiment results showed phosphate absorption of metal-modified biochar materials was stronger than that of unmodified biochar in terms of molecular levels, proving that the modified metal in the biochar structure plays a leading role in H_2_PO_4_^−^ adsorption. Li et al. [21] found the phosphorus adsorption capacity of a nitric acide-Fe^3+^ modified sludge-based biochar was 9.79 mg/g, almost 40 times higher that unmodified biochar.

The response surface method (RSM) is a multivariate technique for statistical and mathematical optimization of process parameters using a minimal number of experimental trials, as well as to analyze the interaction between these parameters [22]. In previous studies, RSM has been frequently applied to describe the effects of biochar preparation conditions, including, heating temperature, holding time, heating efficiency, gas type and gas flow, on the pollutant adsorption capacities of biochars [23,24]. Among these, the heating temperature and holding time significantly affect the pollutant removal efficacy of the biochar [25]. However, little attention has been paid to the optimization of the adsorption conditions using RSM.

In this study, Fe-B composites were synthesized via pyrolysis of three raw biomass materials, following impregnation with FeCl_3_ at different Fe/Biochar mass ratios (0, 0.112, 0.224, 0.448, 0.560, 0.672, and 0.896). The different amounts of modified biochars were added into phosphorus solutions with different pH and initial concentrations to remove and recover phosphorus from wastewater. The specific objectives of this study are as follows: (1) Determine the best Fe/biochar mass ratio for phosphorus removal from water. (2) Compare and optimize the biochar adsorption environmental conditions variables such as pH, ambient temperature, and initial phosphate concentration. The interaction effects of these different parameters on the adsorption performance of the biochar composites were also investigated by RSM. (3) Phosphorus adsorption was carried out under the optimized environmental conditions using the best Fe/biochar mass ratio to clarify the adsorption mechanism using various adsorption isotherms, including Langmuir, Freundlich, and Temkin isotherms.

## 2. Results and Discussion

### 2.1. Phosphorus Removal Efficiency of Fe-B

Previous studies have shown that biochar surface modification not only affects the pollutant removal capacity of biochar, but also influences the pH of the surrounding environment [26]. Therefore, our first step was to investigate the effect of modification with different Fe/C ratios on the phosphorus removal efficiency and solution pH based on a 24 h adsorption experiment. All the raw biochar samples showed low phosphorus removal efficiencies (10.24, 54.32, and 76.48mg kg^−1^ for P, S, and R, respectively). However, the highest phosphorus adsorption capacities of MP, MS, and MR were 823.48 ± 8.40mg kg^−1^ (Fe/C = 0.672), 1079.50 ± 21.50mg kg^−1^ (Fe/C = 1.120), and 1789.78 ± 8.78mg kg^−1^ (Fe/C = 0.672), which were 80.52, 19.87, and 23.40 times the values of the corresponding unmodified biochars, respectively. The possible mechanism is that Fe^3+^ formed complexes with surface functional groups during the modification, resulting in a decrease in the negative charge on the biochar surface and an increase in negatively charged phosphate ions adsorption [27]. Additionally, metal–phosphate precipitation could form via the covalent bonds between the PO_4_^3−^ ion in solution and the metal oxide or cationic compound on the biochar surface, resulting in irreversible adsorption [28].

Biochar pH influences its application in water or soil remediation. This is probably because it affects the ions’ dissolution. It has been observed that a subtle change in soil pH may have an apparent impact on the phosphorus availability. Therefore, biochar in agricultural soil should be applied carefully to avoid unexpected pH variations. The solution pH varied from 7.84 to 9.6, and decreased slightly with the addition of modified biochars compared with unmodified biochars (Figure 1). Further, it gradually decreased with the increase in the Fe/C ratio. The influence extent strongly depended on the Fe content of the biochars, which could be attributed to the acidic features of iron ions [29,30]. In summary, MR_0.672_, MP_0.672_, and MS_0.560_ at Fe/C ratios of 0.672, 0.672, and 0.560, respectively, showed highest PO_4_^3−^ adsorption capacities and had the least impact on solution pH. Therefore, the Fe-B materials have great potential as phosphorus adsorbents for the recovery of phosphorus from aqueous solutions. The phosphorus removal capacity of Fe-B varies with different raw materials, i.e., MR > MS > MP. MR_0.672_ showed the highest phosphorus removal capacity, which was 2.17 and 1.77 times that of the MP_0.672_ and MS_0.560_, respectively. It could be that the pore structure of rape straw is destroyed to a greater extent after pyrolysis and carbonization, and the interior and surface become coarser, so there are more iron oxides attached (Table 1 and Figure 1), increasing its adsorption capacity.

### 2.2. Properties of Fe-B Composites

BET analysis (Table 1) showed the specific surface areas of Fe-B were around 2.17 to 4.69 times those of the unmodified biochars. This indicated that iron particles were successfully loaded on the surface or into the pores of biochars [31]. The morphological structures by SEM showed the pores of the pristine biochars were distinct, tubular, and unoccupied and the surfaces were loose and filled with elongated pore structures (Figure 2a,c,e). However, after Fe modification, the inerratic pores were destroyed, and the number of pores increased while the pore volume decreased (Figure 2b,d,f). The surface of the Fe-B showed a lamellar structure, and was filled with disordered and irregular pores. Fine impurity precipitates were loaded along the pore channels, which provided more adsorption surfaces for phosphate removal. The pH_PZC_ of biochar before and after modification (3.4 and 6.13, respectively), was close to that found in other articles [32].

The FTIR spectra showed similar functional groups between the unmodified and corresponding Fe-B (Figure 3). The broad peaks at 2949–3191cm^−1^ could be attributed to the presence of -CH_2_ groups [33], while the small bands at 2865–3050cm^−1^ indicated the presence of -CH_3_ groups. Further, the peaks at 1388–1430cm^−1^ corresponded to the COO- functional groups and the visible spectral band with a peak at 1075–1085cm^−1^ was contributed by the presence of C-O groups. The broad band in MS, at 3400–3700cm^−1^, indicated the existence of -OH groups enhanced and after modification compared to unmodified biochars, confirming the existence of Fe-OH on the surface of Fe-B [34]. The small bands at 2985–2901cm^−1^ represented C-H stretching vibrations, while the peak at 1560–1665cm^−1^ could be attributed to aromatic C=C stretching vibrations. The peak at 874–833cm^−1^ could be attributed to the bending vibrations of C-H outside an aromatic ring, and the increased projection in the 663–776cm^−1^ fingerprint area could be attributed to the presence of iron oxide, implying the impregnation of iron oxide was successful [35]. The iron oxide possibly originated from the hydrolysis of Fe^3+^ after FeCl_3_ pretreatment, or chelation of ion and oxygen-containing functional groups by double tooth bridges before pyrolysis on the biochar surface [36]. The stripes at 449cm^−1^ could be attributed to the C-Cl vibration [37].

The XRD patterns showed that the 2*θ* values of MR, MP, and MS, were 20.67°, 21.56°, and 20.79°, respectively, with no sharp crystalline fiber peaks, indicating that cellulose decomposed and amorphous structures formed after pyrolysis [38] (Figure 4). The characteristic peaks at 23.81°, 23.35°, and 24.01° were attributed to the (100) crystal plane, indicating the presence of iron oxides or hydroxides in the Fe-B, which again demonstrated Fe was successfully loaded onto the biochar surface. Additionally, a series of diffraction peaks with weak intensities were also observed at 2*θ* values of 35.89°, 36.62°, and 45.52°. This could be the Fe_2_O_3_ dehydrated from the Fe(OH)_3_, which was hydrolyzed to from a small amount of the aqueous FeCl_3_ [39]. The newly generated metal oxides possibly existed as Fe_x_O_y_ particles under weak alkaline conditions, which could then effectively attract negatively charged H_2_PO_4_^−^ and HPO_4_^2−^.

### 2.3. Optimization of P Removal Efficiency

The relationships between environmental variables (including ambient temperature, initial phosphate concentration, and solution pH) and the PO_4_^3−^ removal efficiency of the Fe-B were determined via statistical analysis based on RSM. According to the experimental conditions, the removal efficiency of the biochars varied from 83.43% to 97.20% (Table 2). Analytical methods including linear, interactive, quadratic, and cubic analysis were performed to determine the appropriate type of regression model that sufficiently reflected the above-mentioned correlation. Additionally, the fitting formulas for the PO_4_^3−^ removal efficiencies of Fe-B were determined. 

*Y_MS_*(%) = 68.117 + 0.43 *X_*1*_* − 0.63 *X_*2*_* + 0.042 *X_*3*_* − 0.0003 *X_*1*_X_*2*_* − 0.002 *X_*1*_X_*3*_* + 0.01 *X_*2*_X_*3*_* − 0.001 *X_*1*_*^2^ + 0.02 *X_*2*_*^2^ + 0.004 *X_*3*_*^2^ (R^2^ = 0.994, R^2^(adj) = 0.986, R^2^(pred) = 0.943).

*Y_MP_*(%) = 61.415 + 0.46 *X_*1*_* − 0.25 *X_*2*_* + 0.141 *X_*3*_* − 0.002 *X_*1*_X_*2*_* − 0.003 *X_*1*_X_*3*_* + 0.007 *X_*2*_X_*3*_* − 0.001 *X_*1*_*^2^ − 0.002 *X_*2*_*^2^ + 0.003 *X_*3*_*^2^ (R^2^ = 0.999, R^2^(adj) = 0.999, and R^2^(pred) = 0.993).

*Y_MR_*(%) = 60.565 + 0.46 *X_*1*_* − 0.42 *X_*2*_* + 0.03 *X_*3*_* − 0.005 *X_*1*_X_*2*_* − 0.014 *X_*1*_X_*3*_* + 0.021 *X_*2*_X_*3*_* − 0.001 *X_*1*_^*2*^* − 0.032 *X_*2*_^*2*^* + 0.002 *X_*3*_^*2*^* MR (R^2^ = 0.987, R^2^(adj) = 0.973, and R^2^(pred) = 0.941).

Thus, it is indicated that the model fit well with the experimental data. These R^2^ values indicated that the phosphate removal efficiencies of the biochars were influenced by environmental variables significantly.

### 2.4. Influence of Environmental Factors on Phosphorus Removal Efficiency

The three-dimensional response surface diagram showing the effect of three independent environmental variables (solution pH, ambient temperature, and initial PO_4_^3−^ concentration) and their interactions on phosphorus removal efficiency are presented in Figure 5d–l. The established model shows a significant correlation between the experimental values and the predicted ones (Figure 5a–c).

#### 2.4.1. Effect of pH on Phosphorus Removal Efficiency

pH plays a key role in the phosphorus adsorption process of biochar given that it affects not only the functional groups on the biochar surface, but also the surface charge and the ionization degree of the pollutants on the active site of the adsorbent [40]. Specifically, pH primarily affects phosphorus adsorption by affecting the form of phosphorus, ion exchange, and the availability of competitive adsorption sites. The correlation coefficients of pH on the Fe-B performance of the biochar were −0.63, −0.25, and −0.42 for MS, MP, and MR, respectively (Figure 5). The PO_4_^3−^ removal efficiency increased at first, reaching a maximum of 93.17~94.15%, and then decreased when pH increased. The highest PO_4_^3−^ removal efficiency was observed at pH 6–7. For MS, the PO_4_^3−^ removal efficiency increased from 92.83% (pH = 3) to 96.92% (pH = 7), and then decreased to 85.16% (pH =11). Similar results were also reported for other Fe-based and metal-containing sorbents [41]. This could be because a part of the Fe ions on the biochar surface dissolved under acidic conditions, resulting in a low adsorption capacity of Fe-B [42]. As the solution pH increased to 7, the Fe-B became protonated. As a result, the positively charged Fe-B surface could adsorb the negatively charged phosphate (HPO_4_^2−^ and H_2_PO_4_^−^) owing to a stronger electrostatic attraction between the biochar surface and the phosphate species [43]. As pH increased (7–11), the negative charge on the Fe-B surface strongly repelled the predominant phosphates (HPO_4_^2−^ and H_2_PO_4_^−^), resulting in a decrease in phosphorus adsorption capacity [44].

#### 2.4.2. Effect of Initial Phosphate Concentration on Phosphorus Removal Efficiency

Our experimental results showed that the initial concentrations of phosphate (P_0_) had similar effects on the phosphorus removal efficacies of all the Fe-B. Specifically, the phosphorus removal efficiency increased with the increase of P_0_. A plausible explanation is that the strength of the adsorption driving force increased with increasing phosphate concentration. When the P_0_ was low, the phosphate ions diffused slowly, thus the adsorption reached saturation in a longer time. Because the attachment sites of the biochar were oversaturated with the increase in PO_4_^3−^ concentration, adsorption was gradually inhibited, leading to a decrease in the PO_4_^3−^ removal efficiency.

#### 2.4.3. Effect of Ambient Temperature on P Removal efficiency

A positive correlation was observed between ambient temperature and the PO_4_^3−^ removal efficiency of all the Fe-B. Therefore, it could be deduced that the biochar adsorbtion process is endothermic. On the one hand, the higher temperature, the stronger ion diffusion and interaction forces between the phosphate anions and the biochar surfaces. Consequently, the phosphate could penetrate the internal layered structures of biochar and be easily adsorbed [45]. On the other hand, with a higher temperature, there could be bare agglomeration and deposition of Fe particles on the biochar surface. Thus, ultrafine Fe particles could be sufficiently wrapped on the biochar surface to form a well-layered structure. Additionally, the random thermal motion of ions became stronger at higher temperatures, increasing the possibility of collision between the adsorption sites and the phosphate ions [46].

### 2.5. Optimization of Fe-B via RSM

The maximum experimental phosphorus removal efficiencies of MS, MP, and MR were 97.15, 93.59, and 96.47%, respectively, while the predicted ones were 97.76, 90.23%, and 86.23% under the optimized conditions. This indicated that the statistical results based on BBD were in good agreement with the experimental values. The optimal adsorption conditions for phosphorus removal of both MS and MP were pH 7.0, initial phosphate concentration 132.64mg L^−1^, and ambient temperature 25 °C. Similar results were achieved for Fe/Al (Hydr)oxides-biochar. The maximum PO_4_^3−^ removal capacities of the modified biochars (manure, almond shell, and straw) were significantly increased by 14.05, 25.35, and 21.72 times the values of pristine biochar, respectively [47,48], compared the PO_4_^3−^ removal capacity of modified sludge biochars impregnated with Mg, Ca, Al, Cu, and Fe. The optimum conditions of the pyrolysis process were determined using RSM, indicating that the Ca-rich biochar had the superior adsorption. The PO_4_^3−^ removal efficiency was 85%, which should have resulted from the chemical reaction of CaO and MgO nanoparticles produced during the modified biochar synthesis process and the physical sorption of mesoporous structures.

### 2.6. Adsorption Isotherm

In this study, three equilibrium isotherm models, Langmuir, Freundlich, and Temkin, were applied to describe the phosphate adsorption process. Relevant parameters are listed in Table 3.

The high “goodness of fit” value in the Langmuir and Freunlich adsorption isotherms were higher than those based on Temkin adsorption isotherm, suggesting the phosphate adsorption by Fe-B could be well explained by the Langmuir and Freunlich adsorption isotherms. Correlation coefficients R^2^ of the Langumir and Freunlich adsorption isotherms for Fe-B were higher than 0.95, which can both fit the adsorption process. In the Freunlich model, R^2^ was higher, and the adsorption coefficients 1/n are 0.61823, 0.55680, and 0.46379, respectively, which was favorable adsorption [49]. In addition, the ideal maximum adsorption capacities of the three modified biochars are 3807.99mg kg^−1^, 4560.34 mg kg^−1^, and 5110.81 mg kg^−1^, respectively, according to the fitting results of Langumir model. Thus, we speculated the phosphate was adsorbed primarily by monolayer adsorption on the homogeneous biochar surface. There was no interaction between the adsorbate particles and the adsorption activation energy was uniform. Similar to heterogeneous catalytic reactions, the limiting reaction step was also a surface reaction. As the adsorption energy was constant, there was no surface adsorbate migration. The separation factor or equilibrium constant, R_L_ (a dimensionless constant) increased from 0.34 to 0.93, thus it could be considered that the PO_4_^3−^ adsorption was favorable.

The separation factor or equilibrium constant can be defined as **R***_L_*.
(1)$$R_{L}=\frac{1}{\left(1+K_{L}C_{0}\right)}$$

*C*_0_ represents the initial phosphate concentration. **R***_L_* represents the adsorption equilibrium constant (**R***_L_* values > 1 indicate unfavorable adsorption, **R***_L_* = 1 implies linear adsorption, 0 < **R***_L_* < 1 represents favorable adsorption, and **R***_L_* = 0 represents irreversible adsorption). The **R***_L_* values of all the Fe-B were less than 1, indicating favorable adsorption.

### 2.7. Phosphorus Adsorption Kinetics

It is evident that PO_4_^3−^ was removed spontaneously within the first few minutes, and the removal process reached equilibrium in 10 h (Figure 6). The PO_4_^3−^ removal process was explained using a pseudo-first-order kinetic model as well as pseudo-second-order kinetic model (Table 3). However, the pseudo-second-order kinetic model showed a better fit for the phosphate adsorption process (R^2^ > 0.85). Additionally, the reaction rate was found to be proportional to the number of adsorption sites on the biochar surface. This indicated that, in accordance with [50], the chemisorption of the adsorbate on the biochar was the rate-limiting step for the phosphate removal by MP, MS, and MR.

### 2.8. Mechanism for Phosphorus Adsorption

The mechanisms of phosphate adsorption on biochar surface include ion exchange and electrostatic adsorption. For our Fe-B, these processes primarily depend on solution pH. When solution pH < zero-charge electricity of biochar(pH_pzc_), the biochar surface was positively charged, thus it could electrostatically adsorb H_2_PO_4_^−^ and HPO_4_^2−^. At pH 3–5, the phosphate adsorption capacity of the Fe-B increased. However, at pH 7–9, the adsorption capacities gradually decreased [51]. Precipitation of Fe on biochar surface could be hydrolyzed, and thereafter precipitate as FePO_4_•H_2_O with PO_4_^3−^ [52]. Further, as ligand exchange occurs, a certain number of Fe ions on the biochar surface could combine with phosphate and oxygen-containing functional groups (such as -OH) via bridge bonds. The Fourier spectrum [43] confirmed that Fe-B could bond with phosphate anions by -OH in the amorphous region, contributing to phosphate removal. Our adsorption experiment showed the PO_4_^3−^ was adsorbed onto the biochar surface in a monolayer via electrostatic adsorption or ion exchange. The Fe ions and organic functional groups on the modified biochar surface were mainly responsible for these adsorption processes.

### 2.9. Desorption and Regeneration

To evaluate the application value of the prepared biochars, we investigated their regeneration abilities when they were saturated with the pollutant. The results indicated that the desorption properties of the adsorbents were significantly different (Figure 7) After four cycles of adsorption–desorption, the PO_4_^3−^ removal efficiencies of MP, MS, and MR deceased from 75.10, 75.06, and 90.94% to 36.12, 25.33, and 37.19%, respectively. After the first two cycles, the acidic medium considerably enhanced the desorption efficiency, which could be attributed to the widespread distribution of negative charges on the material surface. Thus, the phosphate became attached to some positively charged functional groups and could be easily removed by the H^+^ ions, which exerted a strong attractive force [53]. After the third cycle, the phosphate adsorption capacity decreased significantly and remained at approximately 75% thereafter. This could be attributed to the inevitable Fe ion leaching or the permanent occupation of active sites after the first adsorption cycle [54]. With the increase of cycles, the decline of adsorbent performance was possibly due to incomplete desorption, changes in adsorbent characteristics (e.g., porosity, crystallinity, and viscosity surface area, and surface precipitation) during desorption and regeneration, and loss of active sites due to material wear [55]. Therefore, the Fe-B had high regeneration potentials, approximately 50% of maximum adsorption capacity after three cycles, and could extensively be used for PO_4_^3−^ recovery and recycling.

## 3. Materials and Methods

### 3.1. Preparation and Characterization of the Biochars 

Soybean straw, rape straw, and peanut shell, obtained from the planting base of Anhui Agricultural University, were selected as raw materials to produce biochar. For the unmodified biochars, three feedstocks were air-dried, grinded to 2mm through a solid waste shredding machine (Hongze XA-1, Hongzen, China), then pyrolyzed in a tubular furnace (Kejin OTF-1200X, Kejin, China) at 600 °C (heating rate 10 °C min^−1^) for 2h in N_2_ atmosphere [56]. Unmodified biochar was labelled as S (for soybean straw), R (for rape straw), and P (for peanut shell) and saved in an airtight container before use. For the Fe-B, a certain amount of unmodified biochars were immersed in 20mL of 10mg L^−1^ FeCl_3_, with the biochar FeCl_3_ solution ratio (*w*/*v*) 0, 0.112, 0.224, 0.448, 0.560, 0.672, and 0.896, respectively. The solutions were adjusted to pH 7.0 by adding 0.1 M NaOH, vibrated at 800 rpm for 24 h through a stable temperature horizontal bath (Langyue HSZ-82A, Langyue, China) and left to stand at room temperature for 24 h. The supernatant was removed and residues were then washed with deionized water to remove residual salts and loosen the attached minerals, then dried at 60 °C for 12 h with suspensions removed. The obtained products in different biochar FeCl_3_ ratios were denoted as MS (for soybean straw), MR (for rape straw), and MP (for peanut shell), and a suffixed number was used if necessary to show the biochar FeCl_3_ ratio, a suffixed b is used to represent the modified biochar with the highest adsorption capacity.

#### 3.1.1. Reagents and Chemicals

FeCl_3_, KH_2_PO_4_, HCl, NaOH, and NaCl were purchased from Hushi Chemical Co., Ltd. (Shanghai, China), these reagents were analytically pure (>99.7%).

#### 3.1.2. Instruments

The specific surface area and pore size of biochars were identified using the Brunauer–Emmett–Teller device (BET, ASAP2460 aperture analyzer, China). Biochar crystalline structure was determined by X-ray diffraction (XRD, Bruker D8 Advance diffractometer, Germany). Biochar surface configuration was observed by scanning electron microscopy-energy dispersive X-ray spectroscopy (SEM-EDX, GeminiSEM 300, Germany). Biochar surface functional groups were identified by Fourier transform infrared spectroscopy (FTIR, iS50FT-IR, Nicolet, USA). A solid sample crusher (VRera XA, XFK, ChangZhou, China) was used for grinding the sorbent, and a mortar was used for granular treatment.

### 3.2. Batch Adsorption Experiments

#### 3.2.1. Phosphate Adsorption Experiments

Batch adsorption experiments examined the adsorption efficiency of different modified biochars for removing phosphorus from aqueous solutions. Initially, 0.3g of each biochar was added in 30 mL (PO_4_^3−^-P 10mg L^−1^) KH_2_PO_4_ solution was of analytical grade (>99.7%). The initial pH for each solution was adjusted to 7.0 before adsorbents addition and the samples were placed in a shaker operating at 200 rpm and 25 °C. After 12 h mixing, the biochar sample mixtures were filtered through a 0.45 μm filter paper (Longjin Membrane Technology Co., Ltd., China) and the concentration of phosphate and pH in the filtrate were analyzed using a UV-Vis spectrophotometer (PERSEE T6, China) at 700 nm wave length and a multiparameter water quality analyzer (pH, Hach HQ30D phc30101, USA), respectively. The experiment was conducted in triplicate, as was the experiment below.

#### 3.2.2. Isotherm Study

The isotherm studies were conducted in 50mL centrifugal tubes with 0.1 g modified biochar and 30mL KH_2_PO_4_ solution (pH = 7.0) of increasing initial concentrations (2, 5, 10, 20, 30, 40, 50 mg L^−1^). The centrifugal tubes were placed in a thermostat shaker (25 °C, 200 rpm) for 24 h. The adsorption capacity of different biochars at equilibrium, *q_e_* (amount of PO_4_^3−^ removed, mg g^−1^) was determined using Equation (3). The PO_4_^3−^ removal efficiency was calculated using Equation (2).
(2)$$\mathrm{Removal}\;\mathrm{efficiency}\;(\%)=\left(1-\frac{C_{e}}{C_{0}}\right)\times 100\%$$
(3)$$q_{e}=\frac{\;\left(C_{0}-C_{e}\right)V\;}{m}$$
where *C_e_* and *C*_0_ (mg L^−1^) referred to equilibrium and initial concentration of PO_4_^3−^; *V* (L) and *m* (g) referred to the solution volume and the sorbent mass.

The models of Langmuir (Equation (4)), Freundlich (Equation (5)), and Temkin (Equation (6)) isotherms were chosen to analyze the adsorption process:(4)$$\mathrm{Langmuir}\;\mathrm{model}:\;q_{e}=\frac{K_{L}q_{m}C_{e}\;}{1+K_{L}C_{e}}$$
(5)$$\mathrm{Freundlich}\;\mathrm{model}:\;q_{e}={\;K}_{f}C_{e}^{1/n}$$
(6)$$\mathrm{Temkin}\;\mathrm{model}:q_{e}=\mathrm{BlnA}+B\ln Ce\;$$
where *q_m_* was the maximum adsorption amount (mg L^−1^), *K_L_* was the Langmuir affinity constant (L mg^−1^), and K*_f_* was the Freundlich constant (mg g^−1^) that signifies the adsorption affinity, *n* was the Freundlich equation exponent related to heterogeneity of adsorbent surface. A was the equilibrium binding constant corresponding to the maximum binding energy, and the Temkin isotherm constant B was related to heat of adsorption (dimensionless).

#### 3.2.3. Kinetic Study

The kinetic study of PO_4_^3−^ removal was conducted in 50mL centrifugal tubes containing 30mL KH_2_PO_4_ solution (PO_4_^3−^ 10mg L^−1^) with 0.1g different biochars. The samples were vibrated at 200 rpm at 25 °C, withdrawn at various time intervals (5, 10, 15, 20, 30 min and 1, 2, 3, 4, 5, 6, 7, 8, 9, 10, 12, 24, 48 h) and filtered with 0.45μm Whatman filter paper. PO_4_^3−^ concentration of leachates was determined by spectrophotometric methods. The amount of PO_4_^3−^ removed by different biochars at time *t* (*qt*, mg (PO_4_^3−^) g^−1^), was calculated using the following equation:(7)$$q_{t}=\frac{\left(C_{0}-C_{t}\right)V}{m}$$
where *C_t_* (mg L^−1^) referred to the PO_4_^3−^ concentration at certain sampling time *t,* V (L) was the solution volume, and m (g) was the sorbent mass.

The pseudo-first-order model (Equation (8)) and pseudo-second-order model (Equation (9)) were applied to analyze the absorption kinetic data.
(8)$$ln\left(q_{e}-q_{t}\right)=lnq_{e}-k_{1}t$$
(9)$$\frac{t}{q_{t}}=\frac{1}{k_{2}q_{e}^{2}}+\frac{t}{q_{e}}$$*k*_1_ is the rate constant of first-order adsorption (min^−1^) and *k*_2_ is the second-order rate constant of adsorption (mg g^−1^ min^−1^). 

### 3.3. Experimental Design Using RSM

Environmental pH, ambient temperature, and initial P concentration were considered as the predominant independent variables, and PO_4_^3−^ removal efficiency (Y, %) was set as the response value. RSM was carried out to analyze the main, interaction, and quadratic effects of three variables on PO_4_^3−^ removal efficiency of MS, MR, and MP. Coded values (−1, 0, +1) of these independent variables were calculated using Box–Behnken design (BBD) (Table 4). The total BBD experiment was constituted of 17 individual runs and the specific test scheme is shown in Table 2. The interaction effect between variables was determined by the second-order quadratic polynomial equation (Version 10.0.6, Stat-Ease Inc., MN, USA), and the predicted response could be quantitatively described as the quadratic function of three variables. The estimated response of P removal efficiency can be expressed as follows:
Y(%) = ω_0_ + ω_1_X_1_ + ω_2_X_2_ + ω_3_X_3_ + ω_12_X_1_X_2_ + ω_13_X_1_X_3_ + ω_23_X_2_X_3_ + ω_11_X_1_^2^ + ω_22_X_2_^2^ + ω_33_X_3_^2^(10)
where X_1_ is initial PO_4_^3−^ concentration, X_2_ is pH, X_3_ is the ambient temperature, ω_0_ is the intercept, ω_1_, ω_2_, and ω_3_ are liner coefficients, ω_12_, ω_13_ and ω_23_ are squared coefficients, and ω_11_, ω_22_, and ω_33_ are quadratic coefficients. The accuracy of the polynomial model was verified by ANOVA. Statistical analysis was performed using Origin Pro 8.0.

### 3.4. Recyclability of Different Modified Biochars

Batch desorption experiments were carried out to study the recyclability of the adsorbents. First, 0.1 g of each saturated adsorbent was added to 30 mL 0.5 M HCl, shaken at 200 rpm at 30 °C for 24 h, and filtered with 0.45 μm Whatman filter paper. The PO_4_^3−^ concentration in the filtrate was determined by UV-Vis spectrophotometry at 700 nm wave length. The desorption cycle was repeated for 4 times after washing the adsorbent with deionized water, and drying at 105 °C. Desorption efficiency (*W_d_*, %) is given as follows:(11)$$W_{d}=\frac{Q_{e}}{Q_{de}}$$
where *Q_e_* is adsorption capacity (mg g^−1^) and *Q_de_* is desorption capacity (mg g^−1^).

## 4. Conclusions

In this study, we investigated the phosphorus removal efficacies of Fe-B derived from three different waste biomass materials and optimized the conditions for their application in phosphorus removal by RSM. Phosphate removal capacities of Fe-B is superior than other unmodified biochar, due to a higher specific surface area and abundant surface functional groups, the increase of adsorption sites, etc. ANOVA showed the phosphorus removal efficiencies of the Fe-B were positively correlated with the initial phosphate concentration and the ambient temperature. The optimal PO_4_^3−^ removal efficiency was 97.71% under the following condition: pH = 7.0, initial PO_4_^3−^ concentration = 132.64 mg L^−1^, ambient temperature = 25 °C. In general, the Fe-B prepared in our study are promising, economical, and environmentally friendly, and are suitable for PO_4_^3−^ removal in water bodies. MR_0.672_, MP_0.672_, and MS_0.560_ showed the best PO_4_^3−^ removal capacity, and MR_0.672_ was the highest among them. 

## Figures and Tables

**Figure 1 molecules-28-02323-f001:**
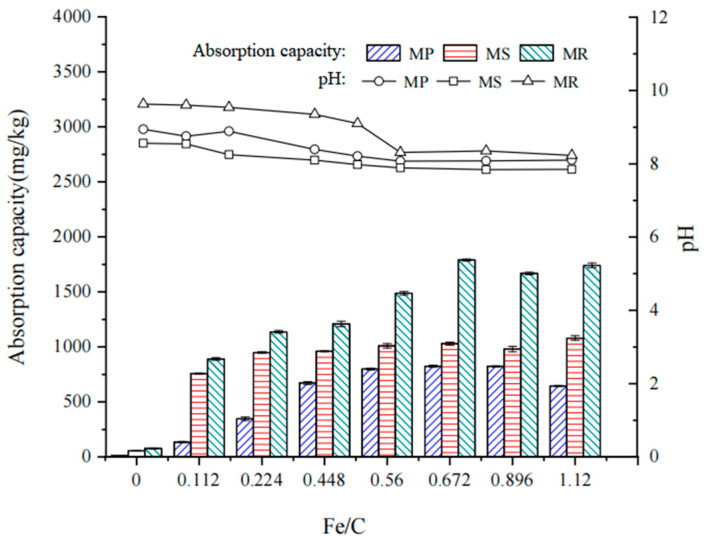
P removal efficiency of Fe-B and pH of the equilibrium solutions.

**Figure 2 molecules-28-02323-f002:**
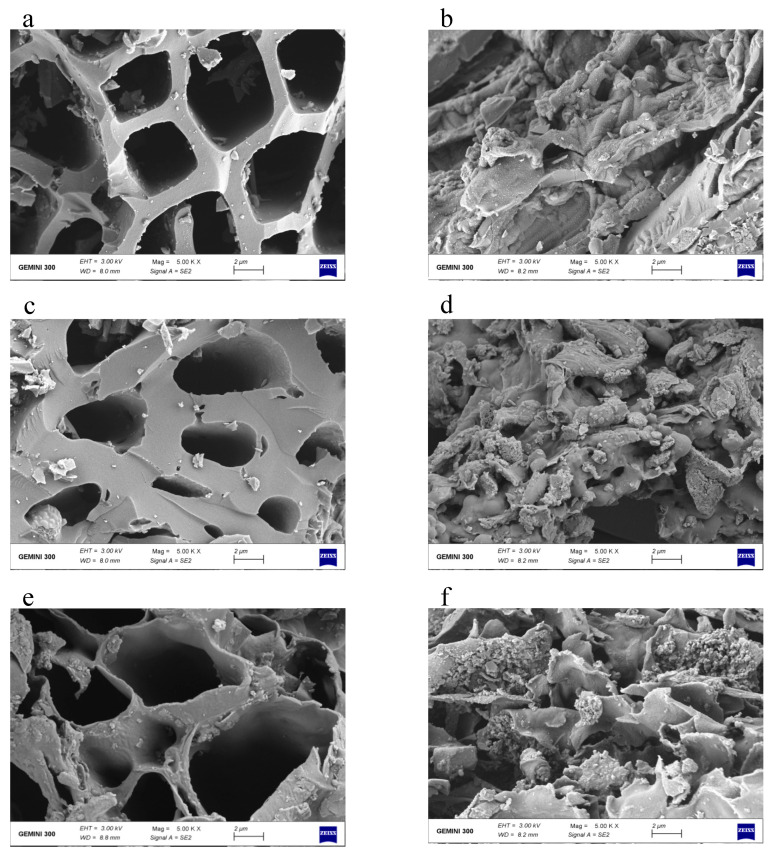
SEM images of unmodified biochars S (**a**), P (**c**), R (**e**) and Fe-B MS (**b**), MP (**d**), and MR (**f**).

**Figure 3 molecules-28-02323-f003:**
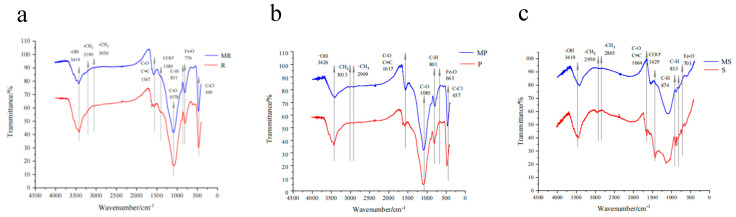
FTIR spectra of the unmodified biochars and MR (**a**), MP (**b**), and MS (**c**).

**Figure 4 molecules-28-02323-f004:**
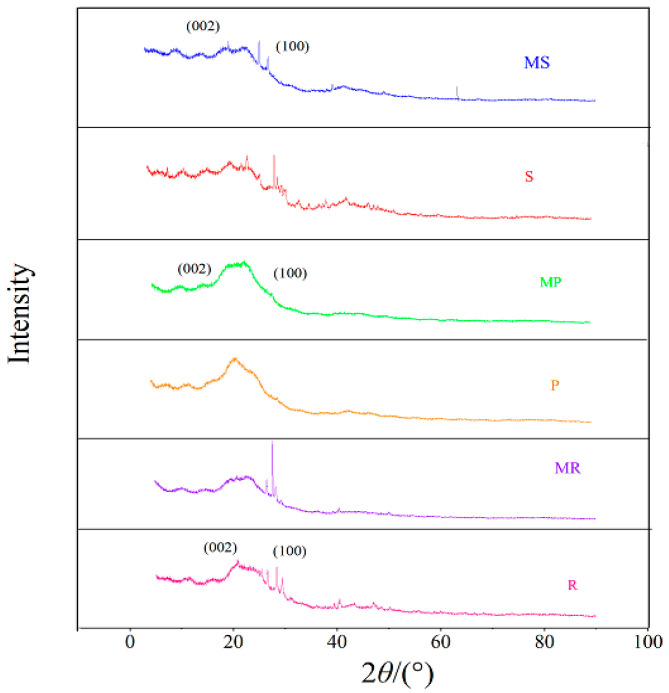
XRD patterns of the pristine and Fe-modified biochar.

**Figure 5 molecules-28-02323-f005:**
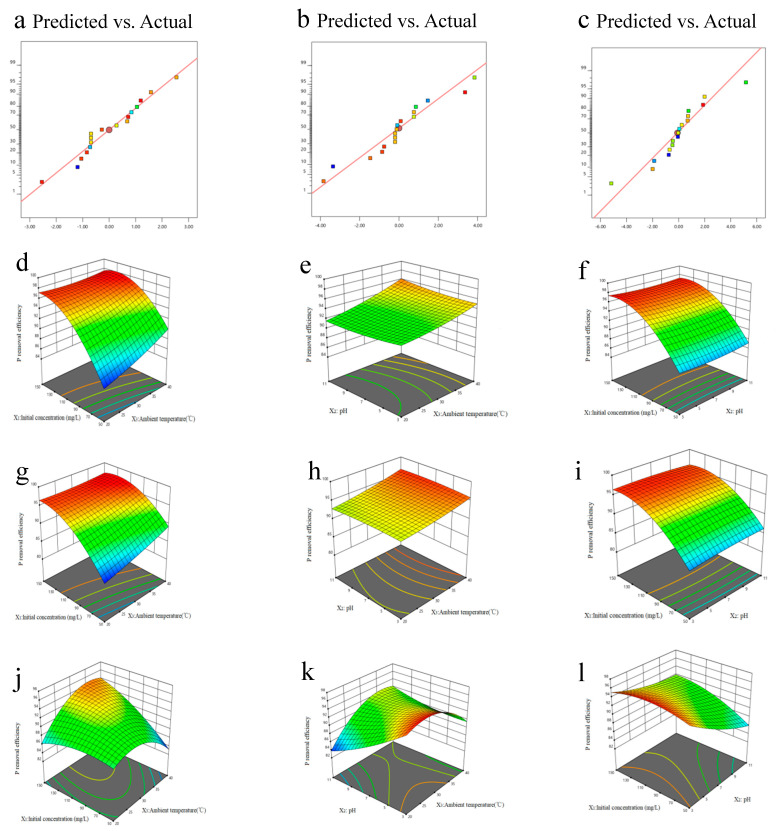
Correlation of the predicted and experimental PO_4_^3-^ removal efficiency of MS (**a**), MP (**b**), and MR (**c**). The 3D surface simulation of C_Initial concentration_ vs. T_Ambient temperature_ (**d**,**g**,**j**), pH vs. T_Ambient temperature_ (**e**,**h**,**k**), and C_Initial concentration_ vs. pH (**f**,**i**,**l**) for the PO_4_^3-^ removal efficiency of MS, MP, and MR, respectively.

**Figure 6 molecules-28-02323-f006:**
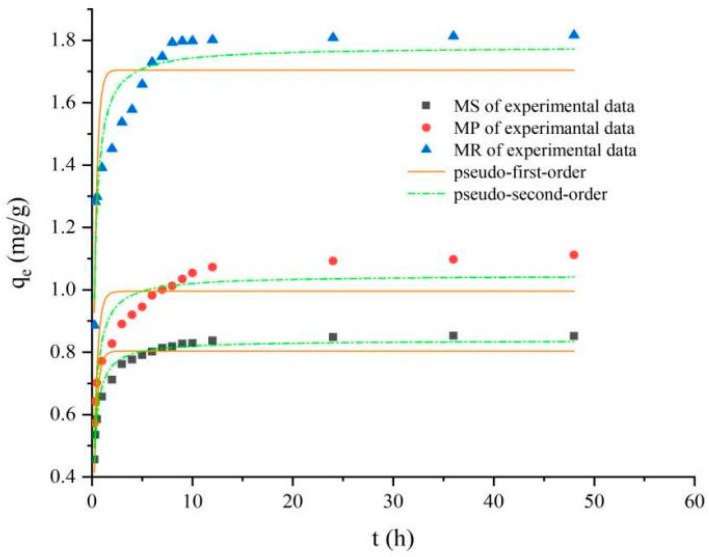
Kinetics of phosphate adsorption on Fe-B fitted by pseudo-first-order and pseudo-second-order models.

**Figure 7 molecules-28-02323-f007:**
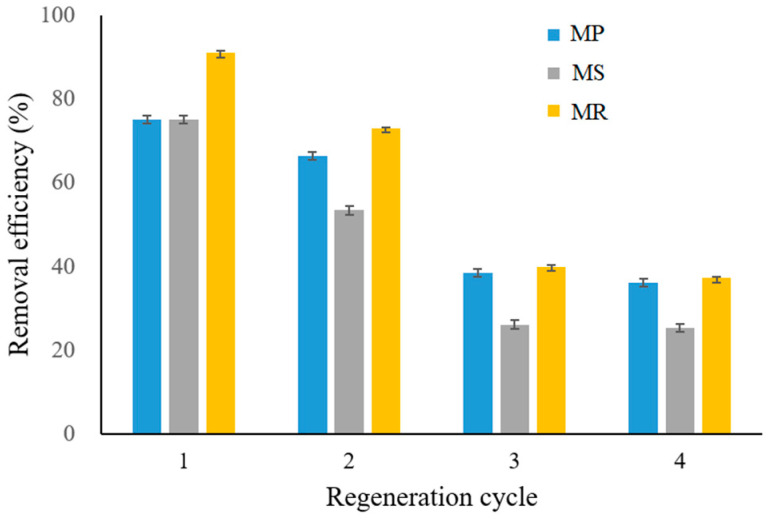
Reusability capacity for MP, MS, and MR.

**Table 1 molecules-28-02323-t001:** Physicochemical properties of biochars before and after modification.

Properties	S	P	R	MS	MP	MR
pH	8.44 ± 0.03	8.83 ± 0.1	9.56 ± 0.3	7.89 ± 0.07	7.89 ± 0.2	8.64 ± 0.3
C%	51.19 ± 1.5	59.19 ± 1.2	77.17 ± 0.6	52.45 ± 1.2	58.77 ± 1.2	65.56 ± 2.0
H%	1.78 ± 1.0	1.44 ± 2.1	1.87 ± 1.5	1.59 ± 1.2	1.36 ± 1.4	1.51 ± 1.4
O%	44.66 ± 1.2	39.4 ± 1.4	31.92 ± 2.1	46.20 ± 1.5	30.07 ± 2.2	20.68 ± 1.7
H/C	0.08	0.07	0.03	0.03	0.02	0.02
O/C	0.85	0.67	0.49	0.90	0.51	0.27
Specific surface areas (g/m^2^)	3.02	7.44	3.05	14.19	16.14	9.43
Fe	-	-	-	0.10	0.14	0.27

**Table 2 molecules-28-02323-t002:** Experimental design and the corresponding results.

Sl.no.	X_1_ Initial Concentration (mg L^−1^)	X_2_ pH	X_3_ Ambient Temperature (°C)	Y_MS_ (%) Experimental	Y_MS_’ (%) Fitted by Model	Y_MP_ (%) Experimental	Y_MP_’ (%) Fitted by Model	Y_MR_ (%) Experimental	Y_MR_’ (%) Fitted by Model
1	100	7	30	94.86 ± 0.05	94.64	94.58 ± 0.04	94.05	93.81 ± 0.06	93.02
2	100	3	20	95.48 ± 0.02	94.84	93.39 ± 0.05	92.83	94.15 ± 0.07	93.79
3	100	11	40	97.39 ± 0.04	96.76	96.47 ± 1.01	95.92	93.44 ± 0.08	92.79
4	50	11	30	87.72 ± 0.11	86.55	86.26 ± 0.04	85.90	85.16 ± 0.03	85.51
5	150	3	30	97.15 ± 0.07	97.12	96.40 ± 0.08	95.65	96.47 ± 0.13	95.33
6	100	7	30	94.86 ± 0.10	94.64	94.59 ± 1.02	94.05	92.19 ± 0.11	92.79
7	50	3	30	87.12 ± 0.05	87.09	85.54 ± 1.05	85.16	93.19 ± 0.06	92.79
8	100	11	20	94.29 ± 0.06	93.35	93.02 ± 0.07	92.29	83.17 ± 0.14	82.51
9	100	3	40	96.83 ± 1.12	96.64	95.70 ± 0.06	95.32	90.17 ± 0.06	91.30
10	50	7	40	90.28 ± 0.10	89.45	89.35 ± 0.05	88.95	83.18 ± 0.07	82.71
11	100	7	30	94.86 ± 0.05	94.64	94.58 ± 0.05	94.05	92.78 ± 0.08	92.79
12	100	7	30	95.78 ± 0.07	94.64	94.58 ± 0.07	94.05	93.17 ± 0.13	92.86
13	150	11	30	97.46 ± 0.04	96.29	95.69 ± 0.05	94.96	94.16 ± 0.06	94.82
14	50	7	20	84.58 ± 0.05	84.42	83.60 ± 0.01	83.43	90.16 ± 0.09	90.47
15	100	7	30	95.47 ± 1.02	94.64	94.58 ± 0.06	94.05	92.16 ± 0.07	92.63
16	150	7	20	96.94 ± 0.07	96.73	96.38 ± 1.02	95.66	86.36 ± 0.07	87.14
17	150	7	40	97.80 ± 0.06	96.91	97.20 ± 0.07	96.26	92.37 ± 0.08	92.79

**Table 3 molecules-28-02323-t003:** Isotherm parameters and kinetic parameters for Fe-modified biochar.

Isotherms	Parameters	MS	MP	MR
Langmuir	*K_L_* (L mg^−1^)	0.03736	0.0429	0.11275
	*Qm* (mg kg^−1^)	3807.99	4560.34	5110.81
	R^2^	0.9919	0.95325	0.97762
Freundlich	*KF*	248.83883	378.87182	853.17842
	*1/n*	0.61823	0.55680	0.46379
	*R* ^2^	0.99624	0.98104	0.98646
Temkin	*A*	0.68666	2.70744	1.93992
	*B*	630.22108	502.93638	925.75533
	*R* ^2^	0.93611	0.74858	0.94703
Kinetics parameters	*Q_e_* (mg kg^−1^)	803.86	995.17	1704.66
Pseudo-first-order	*K_1_* (L mg^−1^)	2.913	2.733	3.140
	*R* ^2^	0.85	0.72	0.75
Pseudo-second-order	*Q_e_* (mg kg^−1^)	837.95	1046.44	1779.2479
	*K_2_* (L mg^−1^)	0.005	0.004	0.003
	*R* ^2^	0.97	0.89	0.88

**Table 4 molecules-28-02323-t004:** Experimental factors and level used in the BBD.

Variable	Symbols	−1	0	+1
Initial concentration (mg L^−1^)	X_1_	50	100	150
pH	X_2_	3	7	11
Ambient temperature (°C)	X_3_	20	30	40

## Data Availability

All relevant data included in the article.

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
