# Peer review of "The Study of Optimal Adsorption Conditions of Phosphate on Fe-Modified Biochar by Response Surface Methodology"

_molecules, 2023, doi:10.3390/molecules28052323_

Round 1
Reviewer 1 Report
In this work an study of optimal adsorption conditions of phosphate on Fe-modified biochar by response surface methodology. The manuscript is very interesting and suitable to be published in this journal, however some major points should be addressed before publication.
1. In introduction please, improve the literature overview on adsorption of phosphore.
2. Please, specify if phosphore was of analytical grade.
3. Characterization of the material should be improved by focusing on the main properties of adsorptive material.
4. Please, specify if pH was monitored during adsorption tests.
5. Please, specify if the zero-charge pH was determined.
6. Please, specify if investigations were carried out in duplicate/triplicate etc.
7. Please, improve comparison between experimental findings and literature data in terms of: adsorption performance, kinetic modelling parameter, isotherm modelling parameter.
8. The maximum adsorption capacities assessed by isotherm seemed to be different with respect to the results from experimental activity reported in Figure 6 : please, clarify this point.
9. Why was it decided to choose Fe as the modifier of biochar?
Author Response
Response to Review Comments:
Manuscript ID molecules-2211795
Title: The study of optimal adsorption conditions of phosphate on Fe-modified biochar by response surface methodology
Authors: Jing Qian, Xiaoyu Zhou, Qingsong Cai, Jinjin Zhao and Xianhuai Huang*
Submitted to: Molecules
Dear Editor, Dear reviewers
Thank you for your letter dated February 9.We were pleased to know that our work was rated as potentially acceptable for publication in journal, subject to adequate revision. We thank the reviewers for the time and effort that they have put into reviewing the previous version of the manuscript. Their suggestions have enabled us to improve our work. Based on the instructions provided in your letter, we uploaded the file of the revised manuscript with all the changes highlighted by using the track change mode in MS word.
Appended to this letter is our point-by-point response to the comments raised by the reviewers.
We would like also to thank you for allowing us to resubmit a revised copy of the manuscript.
We hope that the revised manuscript is accepted for publication in the journal of molecules.
Sincerely,
Qian Jing
Comments from reviewers:
Reviewers #1
In this work an study of optimal adsorption conditions of phosphate on Fe-modified biochar by response surface methodology. The manuscript is very interesting and suitable to be published in this journal, however some major points should be addressed before publication.
- In introduction please, improve the literature overview on adsorption of phosphore.
Authers: Literature overview on adsorption of phosphore was improved as suggusted in line 89-103.
- Please, specify if phosphore was of analytical grade.
Authers: The KH2PO4 used in our experiments was of analytical grade (>99.7%), purchased from Hushi Chemical Co., Ltd. (Shanghai, China). The detailed description was added in lines 146-148 and 162-163.
- Characterization of the material should be improved by focusing on the main properties of adsorptive material.
Authers: We deleted the N% data in Table 3. Remained characterization results are more or less related to the adsorption properties of biochar in lines 286-287.
- Please, specify if pH was monitored during adsorption tests.
Authers: We’ve measured pH of the adsorption solution in a pre-experiment (putted 0.1g biochar into 30mL 10mg/L PO43- solution), but found little difference before and after adsorption. Therefore we did not determine pH in the following experiments.
- Please, specify if the zero-charge pH was determined.
Authers: We have determined the pHPZC of biochar before and after modification (3.4 and 6.13 respectively), it was close to other articles [1], it was added in lines 291-292.
- Please, specify if investigations were carried out in duplicate/triplicate etc.
Authers: Experiments in our manuscript were all conducted in triplicates. This was added in line 169.
- Please, improve comparison between experimental findings and literature data in terms of: adsorption performance, kinetic modelling parameter, isotherm modelling parameter.
Authers: Related content was improved in the manuscript (line 414-431).
- The maximum adsorption capacities assessed by isotherm seemed to be different with respect to the results from experimental activity reported in Figure 6 : please, clarify this point.
Authers: Sorry for the confusion. But Figure 6 was kinetics of phosphate adsorption on Fe-B fitted by pseudo-first order and pseudo second-order models. This was not the same to the maximum adsorption capacity (mg/kg) evaluated by the isotherm.Abdulrah et al.[2] simulated isotherm and kinetic model of strontium asorption in graphere oxide, indicated that The adsorption equilibrium and kinetic results were consistent with the Langmuir isotherm model and the pseudo-second-order kinetic model.When langmuir model parameter qmax was 131.41mg/g, pseudo-second order model parameter qe was 66.050mg/g, the possible reason was the initial concentration were different and the adsorption mechanism of fitting equations were inconsistent.
- Why was it decided to choose Fe as the modifier of biochar?
Authers: It has been reported that metal-modification could provide more active sites, modify the electric charge, and facilitate the contaminants removal capacity of biochar[3,4]. Metal-impregnated biochar could have complex capabilities including redox, adsorption and/or degration[5]. Iron, as one of the most abundant metal in the earth crust, is usually chosen for biochar modification to prepare iron-biochar composites. Fe-impregnated biochar showed high removal efficiency for both cationic and anionic contaminants, and some of them could be separated using an external magnetic field afterwords[6,7].Sun et al.[8] synthesized multifunctional iron-biochar and demonstrated that Fe addition enhanced its removal ability of 1,1,2-TCA and organic matter. Wei et al.[9] synthesized iron loaded sluge biochar to remove two kinds of antibiotics and results showed FOOH/FeO could improve the sorption of tetracycline and doxycycline. However, most of previous studies have focused on the orgainc contaminants removal of Fe-modified biochar, whereas its sorption of phosphorus in water were barely studied.
References
1 Hua,L.L., Huang, X.H., Qian, J., Wang, K. CHaracteristics pf iron modified biochar for nitrogen adsorption in water(In Chinese). China Water & Wastewater. 2022,38(19),61-68.[CrossFef]
2 abdulrahman, A.N., Ahmed, A., Gordan, M. Isotherm and kinetic modeling of strontium adsorption on graphene oxide. Nanomaterials. 2021,11,2780.[CrossFef]
3 Yang, X., Zhang, X.L., Wang, Z.W., Li, S., Zhao. J., Liang, G.W., Xie,X.Y. Mechanistic insights into removal of norfloxacin from water using different natural iron ore-biochar composites: more rich free radicals derived from natural pyrite-biochar composites than hematite-biochar composites. Appl. Catal. B-Environ. 2019, 255, 117752.[CrossFef]
4 Ngambia, A., Ifthikar, I.I., Shahib, A., Jawad, A., Shahzad, M., Zhao, J., Wang, J., Chen, Z., Chen, Z. Asorptive purification of heavy metal contaminated wastewater with sewage sluge derived carbon-supported Mg(II) composite, Sci. Total Environ. 2019,691,306-321.[CrossFef]
5 Cope, C.O., Webster, D.S., Sabatini, D.A. Arsenate adsorptin onto iron oxide amended rice husk char. Sci. Total Environ. 2014, 488:554-561.[CrossFef]
6 Zheng, Y.L., Zimmerman, A.Z., Gao B. Comparative investigation of characteristics and phosphate removal by engineered biochars with different loadings of magnesium, aluminum or iron. Sci. Total. Environ. 2020,747:141277-141293.[CrossFef]
7 Qiu. Y., Xu. X., Xu. Z., Liang, J., Yu, Y., Cao, X., Contribution of different iron species in the iron-biochar composites to sorption and degradation of two days with varying properties. Chem. Eng. J. 2020,389:124771.[CrossFef]
8 Sun, Y., Yu, I.K.M., Tsang, D.C.W., Cao, X., Lin, D., Wang, L., Graham, N.J.D., Alessi, D.S., Komarek, M., Ok, Y.S., Feng, Y., Li, X.D. Multifunctional iron-biochar composites for the removal of potentially toxic elements, inherent cations, and hetero-chloride from hydraulic fracturing wastewater. Tmviron. Int. 2019, 124:521-532.[CrossFef]
9 Wei, J., Liu, J., Li, J., Zhu, Y., Yu, H., Peng, Y. Adsorption and co-adsorption of tetracycline and doxycycline by one-step synthesized iron loaded sludge biochar. Chemosphere. 2019,235, 124254.[CrossFef]
Reviewer 2 Report
line 48: I probably wouldn't use the word "wide" in a sentence in connection with adsorption using biochar.
line 48: "biochar adsorption" is not a technical term, it is adsorption for which biochar is used as an adsorbent
remove from the text "et al." – line 56, 59, 60, 82, 86, 410,
fill/unite the spaces between the unit and the number everywhere in the text, eg: 64; 67; 68; 70; 115; 123; 141; 167 etc.).
line 145-146: The sentence is unnecessary because it is repeated in the following chapter. (equation 1 is missing)
line 78-79: The authors state: „Some biochars contain acidic functional groups on their surface, like straw biochar, which favor cation exchange over anion exchange and show inefficient anion adsorption.“ The truth is that a positively charged surface of the sorbent will be suitable for anion sorption, and a negatively charged surface for anion exchange.
line 152: The autors state: „The P removal efficiency ……“ I recommend correcting and unifying to „PO43-„ values were not converted to phosphorus. (line 165; 166; 168 etc.)
Reminders:
Add in the chapter „2.1. Preparation and Characterization of the biochars“:
What equipment was used for grinding the sorbent, for granular treatment, etc. The name/type of the equipment used and the manufacturer are given.
line 118: Information on the concentration of FeCl3 as well as the purity and manufacturer of the chemical is missing.
line 119: It is not clear why the solution had to be adjusted to pH = 7.0. What was the pH of the untreated solution?
line 120: „….vibrated (better were mixed or shaken) at 800 rpm for 24 h……“ and the specification of the device used is missing
line 127: remove unnecessary characters at the end of the sentence
line 139: It must be clear what the value 10 mg L-1 in parentheses means; add chemical purity, manufacturer, concentration;
line 139-140: It is not clear why the initial pH value for each solution had to be adjusted to 7.0 before adding the adsorbents.
line 141: It is not clear why the phosphate concentration was measured after 12 h.
line 142: add specification of filter paper – manufacturer
line 148-151: The authors state: „The isotherm studies were conducted in 50 mL centrifugal tubes with 0.1 g modified biochar and 50 mL KH2PO4 solution (pH=7.0) of increasing initial concentrations (2,5,10,20,30,40,50 mg L-1). The centrifugal tubes were placed in a thermostat shaker (25 ℃, 200 rpm) for 24 h.“ QUESTIONS: How was it ensured that there was no sample loss at such a high speed (200 rpm)? How was sufficient adsorbent-to-adsorbate contact ensured when a 50-mL vessel containing 50 mL of adsorbate and 0.1 g of adsorbent was used?
In chapter 1 line 109 it is stated that: „Phosphorus adsorption was carried out under the optimized environmental conditions using the best Fe/biochar mass ratio and to clarify the adsorption mechanism using various adsorption isotherms, including Langmuir, Freundlich, and Temkin isotherms.“ but in chapter 2.2.2. Isotherm study the Temkin isotherm is not described.
line 161: the parameter qm is not found in any of the given equations
line 167: The authors state: ...downloaded in different time intervals (5 min to 48 h) - it is better to formulate the text in brackets. It is not clear in what time intervals it was measured. How was the PO43- input concentration controlled?
line 175: error in equation numbering
Serious non-residues
If the filtrate was analyzed by UV spectrophotometry (PERSEE T6, China) at a wavelength of 700 nm, I cannot consider the results reliable. UV spectrophotometry cannot be used to measure this wavelength.
The methodologies are not sufficiently and comprehensibly described.
I miss the discussion in the article.
Author Response
Response to Review Comments:
Manuscript ID molecules-2211795
Title: The study of optimal adsorption conditions of phosphate on Fe-modified biochar by response surface methodology
Authors: Jing Qian, Xiaoyu Zhou, Qingsong Cai, Jinjin Zhao and Xianhuai Huang*
Submitted to: Molecules
Dear Editor, Dear reviewers
Thank you for your letter dated February 9.We were pleased to know that our work was rated as potentially acceptable for publication in journal, subject to adequate revision. We thank the reviewers for the time and effort that they have put into reviewing the previous version of the manuscript. Their suggestions have enabled us to improve our work. Based on the instructions provided in your letter, we uploaded the file of the revised manuscript with all the changes highlighted by using the track change mode in MS word.
Appended to this letter is our point-by-point response to the comments raised by the reviewers.
We would like also to thank you for allowing us to resubmit a revised copy of the manuscript.
We hope that the revised manuscript is accepted for publication in the journal of molecules.
Sincerely,
Qian Jing
Comments from reviewers:
Reviewers #2
line 48: I probably wouldn't use the word "wide" in a sentence in connection with adsorption using biochar.
Authers: “wide” has been changed into “are considered as one of the most promising absorbent materials” in line 48-49.
line 48: "biochar adsorption" is not a technical term, it is adsorption for which biochar is used as an adsorbent
Authers: "biochar adsorption" have been changed into "adsorbent adsorption" in line 48. Line 49 was also improved.
remove from the text "et al." – line 56, 59, 60, 82, 86, 410,
Authers: We removed “et al.” from the text (in line 56, 59, 60, 82, 86, 410 etc.)
fill/unite the spaces between the unit and the number everywhere in the text, eg: 64; 67; 68; 70; 115; 123; 141; 167 etc.).
Authers:We’ve united the number and unit format in the text (line 64, 67, 68, 70, 115, 123, 141, 167 etc.)
line 145-146: The sentence is unnecessary because it is repeated in the following chapter. (equation 1 is missing)
Authers:We deleted this sentence (line 145-146) and added equation 1 in line 176.
line 78-79: The authors state: „Some biochars contain acidic functional groups on their surface, like straw biochar, which favor cation exchange over anion exchange and show inefficient anion adsorption.“ The truth is that a positively charged surface of the sorbent will be suitable for anion sorption, and a negatively charged surface for anion exchange.
Authers:We corrected the sentence about “Some biochars contain acidic functional groups on their surface like straw biochar, which would be suitable for cation adsorption,in line 80. Shaaban et al.[1] suggested that acidic functional groups such as phenol, lactonic and carboxylic acid of pore surface negative charge property contributes to better cation exchange capacity that help to retain cation nitrogen nutrient compound in soil such as ammonium NH4+.
line 152: The autors state: „The P removal efficiency ……“ I recommend correcting and unifying to "PO43-" values were not converted to phosphorus. (line 165; 166; 168 etc.)
Authers:We changed and unified “the P removal efficiency ” and “Phosphorus” into “PO43-” in line 162, 193, 197,198 etc.
Reminders:
Add in the chapter „2.1. Preparation and Characterization of the biochars“:
Authers: the chapter “2.1. Preparation and Characterization of biochars,” in line 127.
What equipment was used for grinding the sorbent, for granular treatment, etc. The name/type of the equipment used and the manufacturer are given.
Authers: The name, type and manufacturer of these equipments were added in “2.1.2 Instruments” (line 138-139 and line 131-132,line 156-157 ).
line 118: Information on the concentration of FeCl3 as well as the purity and manufacturer of the chemical is missing.
Authers:We added the FeCl3 purity was analytical pure (>99.7%), pruchased from Hushi Chemical Co., Ltd. (Shanghai, China) in line 147-148. Its concentration 10mg/L was added in lines 135.
line 119: It is not clear why the solution had to be adjusted to pH = 7.0. What was the pH of the untreated solution?
Authers: We adjusted the solution pH in order to unify the solution pH of FeCl3 with increasing concentration and eliminate the interference of pH on biochar midification. The untreated solution is acidic, with pH around 3-4.
line 120: „….vibrated (better were mixed or shaken) at 800 rpm for 24 h……“ and the specification of the device used is missing
Authers: We added the name and type of the device in line 138-139.
line 127: remove unnecessary characters at the end of the sentence
Authers:We removed unnecessary characters at the end of the sentence.(line 145)
line 139: It must be clear what the value 10 mg L-1 in parentheses means; add chemical purity, manufacturer, concentration;
Authers: The value 10mg L-1 refers to the concentration of in KH2PO4 solution. We added the chemical purity (>99.7%,analytical grade), manufacturer (Hushi Chemical Co., Ltd., Shanghai, China) in lines 162 and 146-147, as well as other chemical reagents.
line 139-140: It is not clear why the initial pH value for each solution had to be adjusted to 7.0 before adding the adsorbents.
Authers: We adjusted the pH of the solution to 7 to eliminate the interference of the solution pH on the results during the adsorption experiment.
line 141: It is not clear why the phosphate concentration was measured after 12 h.
Authers: 12h is enough for the adsorbent biochar contacted and reacted with the phosphate solution fully. According to the kinetics of phosphate adsorption (figure 6), Fe-B adsorption of PO43- reached equilibrium in around 10min .Guo et al.[2] studied the adsorption isotherm of a modified wheat stalk biochar on cadmium and lead with agitation at 200rpm for 6h.Wang et al.[3] studied the adsorption of Fe/Ca modified rice straw biochar on phosphorus with 0.05 g biochar and 25 mL P solution shaking at 150 rpm for 2 h at 20℃.
line 142: add specification of filter paper – manufacturer
Authers: We added pecification of filter paper in line 166.
line 148-151: The authors state: "The isotherm studies were conducted in 50 mL centrifugal tubes with 0.1 g modified biochar and 50 mL KH2PO4 solution (pH=7.0) of increasing initial concentrations (2,5,10,20,30,40,50 mg L-1). The centrifugal tubes were placed in a thermostat shaker (25 ℃, 200 rpm) for 24 h.“ QUESTIONS: How was it ensured that there was no sample loss at such a high speed (200 rpm)? How was sufficient adsorbent-to-adsorbate contact ensured
Authers: We’ve sealed the centrifugal tubes with films and covers tightenly before vibration and monitored the tubes during vibration in order to prevent sample losing. The adsorbent-to-adsorbate ratio and oscillation conditions are frequently used in isotherm study of biochar adsorption. Wang et al.[4] added 0.4g lignin-derived biochar into 8 ml Pb(II) solution and shaked the mixture for 24 hours to study the Pb(II) adsorption isotherms. Srinivasan et al.[5] used 1g groundnut shell-based biochar into 100mL wastewater, and shaked at 200rpm for 6h to attain the greatest sorption capacity.
when a 50-mL vessel containing 50 mL of adsorbate and 0.1 g of adsorbent was used?
Authers:Sorry for the mistake. We corrected it in line 171-172. The isotherm studies were conducted in 50mL centrifugal tubes with 0.1 g modified biochar and 30mL KH2PO4 solution.
In chapter 1 line 109 it is stated that: „Phosphorus adsorption was carried out under the optimized environmental conditions using the best Fe/biochar mass ratio and to clarify the adsorption mechanism using various adsorption isotherms, including Langmuir, Freundlich, and Temkin isotherms.“ but in chapter 2.2.2. Isotherm study the Temkin isotherm is not described.
Authers: We added the description of the Temkin isotherm in chapter 2.2.2, and rearranged the formula number.
line 161: the parameter qm is not found in any of the given equations
Authers: Sorry for the mistake. We corrected the numerator qe to qm in langmuir model in equation 3.
line 167: The authors state: ...downloaded in different time intervals (5 min to 48 h) - it is better to formulate the text in brackets. It is not clear in what time intervals it was measured. How was the PO43- input concentration controlled?
Authers: Different time intervals (5, 10, 15, 20, 30 min and 1, 2, 3, 4, 5, 6, 7, 8, 9, 10, 12, 24, 48h) was added in line 195-196.
line 175: error in equation numbering
Authers:We corrected the equation numbers.(line 204-205).
Serious non-residues
If the filtrate was analyzed by UV spectrophotometry (PERSEE T6, China) at a wavelength of 700 nm, I cannot consider the results reliable. UV spectrophotometry cannot be used to measure this wavelength.
Authers: Ammonium molybdate spectrophotometry method (700 nm) (GB 11893-1989) has been widely used to determine phosphorus concentration in solution. Chang et al.[6] used UV spectrophotometry at 700 nm to determine the total P concentration in water. Carreres-Prieto et al.[7] measured total P concentration from the spectral response of samples by UV spectrophotometry at 700 nm.
The methodologies are not sufficiently and comprehensibly described.
Authers: Materials and methods were improved. We added the 2.1.1 Reagents and chemicals and 2.1.2 Instruments.
I miss the discussion in the article.
Authers: Sorry for the confusion. We put the discussion part and the results together in chapter3,and this part has been improved.
Conferences:
1 Shaaban, A., Se, S.M., Mitan, N.M.M., Dimin, M.F. Characterization of biochar derived from rubber wood sawdust through slow pyrolysis on surface porosities and functional groups.Procedia Engineering.2013,68,365-371.[CrossFef]
2 Gui, D.D. Zhai X.W. Study on the adsorption properties and mechanism of Pb2+ and Cd2+ by modified biochar . Applied Chemical Industry. 2023,1-8.[CrossFef]
3 Wang, R.Z., He, Q., Guo, C.C., Bao, Y.M., He, X., Cao, Y., Li, P., HuangFu, X.L. Adsorption capacity of phosphorus in septic tank by rice straw biochar loaded with Fe/Ca. Journal of Civil and Environmental Engineering. 2022,265,2096-6717.[CrossFef]
4 Wu,F.F., Chen. L., Hu. P., Wang, Y.X., Deng. J., Mi. B.B. Industrial alkali lignin-derived biochar as highly efficient and low-cost adsorption material for Pb(II) from aquatic environment. Bioresource technol. 2021,322,124539.[CrossFef]
5 Srinvasan, K., Bobbili, A.S., Gokulan, R., Balakumar, S., Mureaklikrisknan, R., Janakani, U.B.S.V., Bonu. Ramesh., Nasar, R. Effective removal of reactive yellow 145 (RY145) using biochar derived from groundnut shell. Adv. Mater. Sci. Eng. 2022,2022,1-7.[CrossFef]
6 Cheng, X.L., Huang, Y.N., Li, R., Pu, X.C., Huang, W.D., Yuan, X.F.Impacts of water temperature on phosphorus release of sediments under flowing overlying water. J. Contam. hydrol. 2020, 235,103717.[CrossFef]
7 Carreres-prieto, D., Garcia, J.T., Cerdan-Cartagena, F., Suardiaz-Muro, J., Lardin, C. Implementing Early Warning Systems in WWTP. An investigation with cost-effective LED-VIS spectroscopy-based genetic algorithms. Chemosphere.2022, 293, 1-12.[CrossFef]
Round 2
Reviewer 2 Report
The electromagnetic spectrum of ultraviolet radiation (UVR), is defined most broadly as 10–400 nanometers and visible as 400-700 nm. If I state that I measure at a wavelength of 700 nm, I have to use a spectrophotometer that works in the visible spectrum, not in the UV region. Therefore, it needs to be fixed.
